Performance Evaluation of Multi-Source Methane Emission Quantification Models Using Fixed-Point Continuous Monitoring Systems

David Ball<sup>1</sup>, Umair Ismail<sup>1</sup>, Nathan Eichenlaub<sup>1</sup>, Noah Metzger<sup>1</sup>, and Ali Lashgari<sup>1</sup>

<sup>1</sup>Project Canary, Denver, CO

Correspondence: Ali Lashgari (ali.lashgari@projectcanary.com)

Abstract. Quantifying methane emissions from oil and gas facilities is crucial for emissions management and accurate facility-level GHG inventory development. This paper evaluates the performance of several multi-source methane emission quantification models using the data collected by fixed-point continuous monitoring systems as part of a controlled release experiment. Two dispersion modeling approaches (Gaussian plume, Gaussian puff) and two inversion frameworks (least-squares optimization and Markov-Chain Monte-Carlo) are applied to the measurement data. In addition, a subset of experiments are selected to showcase the application of computational fluid dynamic (CFD) informed calculations for direct solution of the advection-diffusion equation. This solution utilizes a three-dimensional wind field informed by solving the momentum equation with the appropriate external forcing to match on-site wind measurements. Results show that the Puff model, driven by high-frequency wind data, significantly improves localization and reduces bias and error variance compared to the Plume model. The Markov-Chain Monte-Carlo (MCMC) based inversion framework further enhances accuracy over least-squares fitting, with the Puff MCMC approach showing the best performance. The study highlights the importance of long-term integration for accurate total mass emission estimates and the detection of anomalous emission patterns. The findings of this study can help improve emissions management strategies, aid in facility-level emissions risk assessment, and enhance the accuracy of greenhouse gas inventories.

## 15 1 Introduction

Quantification of methane emissions from oil and gas facilities is crucial for facility-level emissions management and accurate greenhouse gas (GHG) inventory development (Sharafutdinov 2024). Understanding the contribution of different emission sources to overall site emissions allows operators to improve asset risk management and prioritize mitigation efforts. Currently, the US Environmental Protection Agency (EPA) and many other entities use bottom-up GHG emissions inventories, which mainly rely on activity rate and emission factors (Allen, Zimmerle, and Dabbar 2024).

Several studies have highlighted major shortcomings of bottom-up inventories (Riddick and Mauzerall 2023; Riddick, Mbua, Anand, et al. 2024; Riddick, Mbua, Santos, et al. 2024). In 2021, 70% of methane emissions came from facilities emitting less than 100 kg/hour, with 30%, 50%, and approximately 80% coming from facilities emitting less than 10 kg/hour, 25 kg/hr, and 200 kg/hour, respectively (Williams et al. 2025), demonstrating that low-emitting facilities, particularly those below the detection limit of most point-source remote sensing platforms, contribute significantly to total oil and gas methane emissions. Therefore, it is essential to employ approaches that accurately account for the substantial impact of these small sources.

The substantial variability in methane emission intensity across geographic regions, facility types, and operators necessitates a comprehensive characterization of emission events. Ideally, by applying methods that enable emission event detection, localization, and quantification (DLQ), the distributions of rates, durations, and frequencies can be inferred to provide a deeper understanding of site-specific emission patterns, and may bring to light any underlying issues, and aid in root cause analysis.

Direct measurement is essential for a more thorough characterization of emission events. A wide variety of methods can be used to collect various data related to methane emissions. Fixed-point continuous monitoring systems (CMS) have been widely deployed to monitor emissions from oil and gas production facilities for several years. These systems were initially deployed as a means for emissions anomaly detection, and as such, were commonly referred to as "smoke alarms" (IJzermans et al. 2024; Gosse 2023). They were intended to provide timely alerting of elevated emissions by processing raw concentration signals into alerts via a variety of anomalous event detection algorithms, ranging from static concentration thresholds to more sophisticated approaches employing signal processing and/or machine learning methods (Gosse 2023). While anomaly detection is a useful function of CMS, providing additional information regarding the source locations and magnitude of emission events would enhance the actionable insights provided by these systems. If these additional features can be developed, validated, and proven to be reliable, some of the key benefits that CMS could offer include (i) providing a comprehensive picture of site-level emissions for the entire period of deployment, (ii) rapid detection of emissions ranging from relatively low rates to super emitting events, (iii) capturing both short-duration/intermittent and continuous events, (iv) accurate time-bounding of intermittent emission events, (v) providing equipment-specific emissions insights that can aid in root cause analysis and provide strategically relevant information for targeted mitigation efforts, and (vi) complimenting other measurement methods using a continuous stream of site-specific data on emission estimates, direct concentration measurements, and meteorology. To expand the application of CMS, it is crucial to improve the existing understanding of its quantification performance, including its accuracy and uncertainty. This will further demonstrate the value of these systems as measurement tools that can not only detect and time-bound anomalous emission events but also provide insight into the total emissions originating from a given facility over time and the contribution of different sources to the facility-level emissions.

Several studies have independently evaluated the efficacy of CMS in quantifying emissions, suggesting promising advancements in recent years (Bell et al. 2023; Ilonze et al. 2024; Cheptonui et al. 2025). Although technologies have demonstrated

50

marked improvements in quantification accuracy, the algorithms behind these results are proprietary, making it difficult to compare the results from different technologies of the same sensing modality. Proprietary DLQ algorithms, while understandable in a competitive business environment, are often seen as black boxes, requiring different, more involved methods to evaluate their uncertainties and ensure that their performance is fully understood across various environments. In addition, evaluating methane emission solutions as a single package (combining measurement, data collection strategy, and data processing) yields an inseparable uncertainty value that reflects the combined uncertainties from all three processes. This makes it impossible to differentiate the uncertainty arising from the measurement (i.e., hardware), data collection (i.e., deployment strategy and sensor configuration), and data processing (the application of DLQ algorithms).

While extensive research exists addressing pollutant transport (including long-range dispersion, localization, and quantification at much larger scales for other applications), (Chen, Modi, et al. 2022; Schade and Gregg 2022; Karion et al. 2019; Peischl et al. 2016) relatively little literature focuses on methane emission quantification using near-source point sensor measurements, i.e., measurements within the boundaries of upstream oil and gas facilities. Most of the literature employs existing dispersion modeling tools and methods, such as AERMOD (Cimorelli et al. 2005) and CALPUFF (Allwine, Dabberdt, and Simmons 1998), or quantify single-source emissions (Sharan et al. 2009; Zhang et al. 2019; Kumar et al. 2022; Daniels, Jia, and Hammerling 2024a; Chen, Schissel, et al. 2023; Chen, Kimura, and Allen 2024).

In a study published in 2019, a steady-state Gaussian plume model was employed to estimate emission rates from point sources (Zhang et al. 2019). In this method, the plume spread parameters are simplified for short distances, and a heuristic dispersion modifier is introduced to account for non-ideal measurement conditions. This quantification method is intended to be used in conjunction with source localization techniques. A more recent study (Daniels, Jia, and Hammerling 2024b) provides a framework (and open-source implementation) for single source detection, localization, and quantification, with promising results for cases where only a single emission source is present. However, the fact that the algorithm only identifies a single source per emission event renders the algorithm inapplicable to general use cases. Comprehensive reviews of advanced detection and quantification methods can be found elsewhere (Hollenbeck, Zulevic, and Chen 2021; Yang et al. 2023).

Dispersion models and inversion frameworks are essential tools for translating ambient methane concentration measurements (e.g., ppm readings) into source flux rates (mass of pollutant emitted per unit of time). Forward-running dispersion models simulate how methane released from a source disperses in the atmosphere based on a number of meteorological variables including wind speed, direction, and atmospheric stability. On the other hand, inversion models use mathematical techniques to estimate the source flux(es) that would have resulted in the observed ambient concentrations at the sensor location. This often involves solving an optimization problem, where the inverse model adjusts the source strengths and locations until the simulated concentrations best match the observed amounts.

This paper aims to address the critical need for developing a more comprehensive understanding of the performance and robustness of various multi-source methane quantification methods by evaluating the performance of several established atmospheric
dispersion modeling and inversion frameworks within a controlled, multi-leak experimental setting with synchronous emission
sources and constant rates. This study leverages data collected from a fixed-point CMS deployed at a simulated oil and gas site
with multiple simultaneous methane releases of varying magnitudes and locations. The accuracy and reliability of these models are evaluated with respect to several key metrics related to localization accuracy and total-facility (i.e., source-integrated)
quantification accuracy.

Three key questions will be addressed in this study: (i) under an optimum sensor density and placement, how effectively can a CMS pinpoint emissions to the correct equipment group? (ii) what is the accuracy of the total site-integrated emissions estimates for such CMS network? And, (iii) How well can an advanced CFD-based forward model, coupled with various inversion frameworks perform in predicting emission rates compared to traditional plume and puff models?




Accurate emissions quantification using CMS can enhance the reliability and robustness of GHG emissions inventory development. Traditional inventory methods often rely on activity data and generic emission factors, which fail to capture the dynamic nature of emissions from individual sources or facilities. By providing continuous, real-time measurements source-specific emissions, CMS offers a direct and empirically driven approach to quantify actual emissions. High temporal resolution of the CMS measurement allows for the identification and characterization of gas releases, including the duration and frequency of emission events. In addition, accurate quantification offers a more in-depth understanding of the magnitude of fugitive emissions, intermittent events, and variations in operational performance that are often missed by periodic or estimation-based methods. Integrating CMS data into GHG inventories leads to a more comprehensive understanding of emission sources, enables the tracking of emission reduction efforts with greater confidence, and supports the development of more granular and verifiable inventories, informing climate policies, and tracking progress towards decarbonization goals.

In this study selected quantification algorithms are evaluated using the data from controlled release experiments featuring constant-rate emission events with known start and end times. However, it's crucial to recognize that these controlled release scenarios are highly idealized, as they involve constant release rates and simultaneous emissions from all active sources. This idealization may impact the practical applicability of these algorithms in more complex, real-world conditions. A more indepth evaluation of the performance of fixed-point CMS in complex emission environments is provided in a separate study (Ball, Eichenlaub, and Lashgari 2025).

This work offers a novel contribution by evaluating several multi-source methane quantification techniques using multi-leak, controlled-release data. Unlike previous studies that often rely on simulations, this study leverages a fixed-point CMS to cap-

ture the complexities of overlapping plumes from simultaneous releases. This approach provides a unique opportunity to assess model accuracy and reliability under semi-realistic field conditions representative of relatively simple upstream oil and gas facilities. By emphasizing the strengths of each technique, this study offers crucial insights for improving methane emission quantification strategies, including guidance for selecting appropriate dispersion models and inversion tools, ultimately informing the development of more effective methane mitigation in the oil and gas industry.

## 2 Data


The data presented in this work are all collected with the Canary X integrated device, which includes a tuneable laser diode spectroscopy (TDLAS) methane sensor and can additionally be mounted with an ultrasonic anemometer (at least one of which is required for the sensor network to perform quantification). The Canary X integrated monitoring devices use TDLAS technology coupled with other necessary components to serve as an IoT-enabled stand-alone monitoring device with high sensitivity to perform high-fidelity measurement of methane concentrations, crucial for accurately quantifying emissions in the field. The methane measurement sensors have 0.4 ppm sensitivity, ±2% accuracy, and a precision of ≤0.125 ppm with 60-second averaging. This integrated measurement device is capable of 1 Hz sampling, although the measured quantities are often aggregated to 1-minute averages for the purposes of analysis and applying quantification algorithms. Throughout this work, a note will be made any time 1Hz data is used for a specific piece of an algorithmic workflow: if not otherwise stated, quantities are assumed to be minute-averaged aggregates.


Methane concentration measurements are complemented with meteorological data collected on-site using RM Young 2D ultrasonic anemometers. Manufacturer specifications indicate the anemometers have an accuracy of  $\pm 2\% \pm 0.3$  m/s for wind speed and  $\pm 2^{\circ}$  for wind direction, with resolutions of 0.01 m/s and 0.1°, respectively. Methane concentrations and meteorological data are continually published to a cloud server using cellular networks.



The data was collected over 82 days (February to April of 2024) as part of an independent, single-blind controlled release study performed at the Colorado State University (CSU) Methane Emission Technology Evaluation Center (METEC) facility in Fort Collins, Colorado. METEC is a research facility hosted by the CSU Energy Institute that facilitates the study of methane leaks from oil and gas infrastructure as part of the Advancing Development of Emissions Detection (ADED) project, funded by the Department of Energy's National Energy Technology Laboratory (NETL). Ten Canary X integrated devices were installed within the METEC site perimeter to measure ambient methane concentrations. All of the Canary X devices used for this study were equipped with an anemometer. More details on the data collected as part of the 2024 CSU METEC controlled release study can be found elsewhere (Cheptonui et al. 2025).

Controlled release "experiments" at the METEC facility include between 1 and 5 releases that are synchronously turned on and off at the start and end of each unique experiment. The rate of each source is held approximately constant during an individual experiment. Experiment durations ranged between 30 minutes and 8 hours while individual source rates ranged from 0.081 to 6.75 kg/hr. Figure 1 overviews the number of active releases per experiment and source release rates. The experiments are designed such that only one release point is active per equipment group at the METEC facility. Each equipment group is composed of numerous "equipment units" (i.e., individual tanks, wellheads, or separators) and each equipment unit may have multiple potential release points on it. In other words, each equipment group has numerous *potential* release points, but only one is ever active at a time for a given experiment. In this study, we focus on the ability of the system to correctly detect, localize, and quantify to the equipment group level. As such, the centroid of each equipment group is computed and these 5 coordinate pairs (corresponding to the 5 equipment groups at the facility) are used as the potential source locations as an input to the localization and quantification (LQ) algorithms. The heights of the release locations are unknown but assumed to be 2 meters tall for all sources except for the group of tanks in the middle of the facility, for which a height of 4.5 meters is assumed and used as input to the LQ algorithms.



**Figure 1.** (a) Histogram of active releases per experiment, and (b) Histogram of source release rates (kg/hr).

Figure 2 offers a visual illustration of the layout of the controlled release facility (left), including bounding boxes around each of the 5 equipment groups (left) and sensor locations (x's). It also shows measurement data from a randomly selected controlled release experiment, including concentration measurements from individual sensors (top right) and the QU and V components of the anemometer measurements (with solid and dotted lines, respectively, bottom right). The colors of the curves in the right panels here correspond to the colored x's in the left panels. This figure encapsulates all of the data necessary to run quantifica-


For the purposes of this study, the known release start and end times are used to segment out the relevant measurement data for each experiment; detecting and time-bounding each unique experiment is outside the scope of this work. Therefore, by applying several selected quantification algorithms to data from individual experiments, this study evaluates how well the quantification algorithms perform on constant-rate emissions events with known start and end times. It's important to note that these controlled release scenarios are highly idealized, featuring constant release rates and simultaneous emissions from all active sources, which may limit the practical application of these algorithms. However, it is still useful to evaluate the efficacy of various quantification algorithms under idealized setups to lay the groundwork for future development and studies in which these underlying assumptions and simplifications will be relaxed to more accurately reflect real-world conditions.

**Figure 2.** Facility layout including source and sensor positioning (left) and example of data taken during a single experiment (top right: concentration measurements, bottom right: wind measurements). The colored boxes in the left panel show the spatial extent of each of the 5 equipment groups. Each color in the concentration and wind speed curves (right panels) corresponds to a colored x representing sensor location in facility layout (left) panel. Sensor locations are shown in a zero-referenced easting/northing projected coordinate system as opposed to latitudes and longitudes so that the relative spatial distances are more visually interpretable.

## 185 3 Methodology

This section, along with the Supplemental Information detail several dispersion models and inversion frameworks as options to quantify methane emissions based on ambient concentration measurements using fixed-point CMS.

## 3.1 Dispersion Models

Two distinct methods, the Gaussian plume and Gaussian puff models, for predicting concentrations at receptor locations given a set of sources and associated rates are detailed in the Supplemental Information. The theoretical and fundamental aspects as well as the underlying assumptions of each method are described and in-depth discussions of various aspects of implementation to establish a robust foundation for their use is offered. In addition to these two most commonly used forward dispersion modeling methods, a more novel approach, a CFD-informed calculation is included in the Supplemental Information, that directly solves the advection-diffusion equation with a three-dimensional wind field informed by solving the momentum equation with the appropriate external forcing to match on-site wind measurements. All of these three methods rely on (or are derived from) the advection-diffusion equation, also commonly referred to as the scalar transport equation. For an incompressible flow with homogeneous and isotropic diffusion, this equation can be written as:

$$\frac{\partial C}{\partial t} + \boldsymbol{u}(\boldsymbol{x}, t) \cdot \nabla C - D\nabla^2 C = Q(\boldsymbol{x}, t)$$
(1)

where C represents the concentration, u is the wind vector (which may vary as a function of both space and time), D is the diffusion coefficient, and Q represents emission sources (which can also vary as a function of both space and time). It is important to note that unless the treatment of the wind field, u, explicitly accounts for chemical buoyancy, the resulting solution of the advection-diffusion equation will not capture this effect.

For the purposes of this study, we will focus on cases with constant emission rates, i.e. Q(x,t) = Q(x). Inversion frameworks to infer time-varying source rates will be addressed in future work. Furthermore, we will focus on emissions from discrete point sources, where the sizes of the orifices of the controlled release systems are of the order  $\sim$  centimeters, much smaller than source-receptor distances of the order  $\sim$ 10 meters. As such, the constant-rate source function Q(x), can be expressed as the summation of discrete point sources of varying rates:

$$Q(\boldsymbol{x}) = \sum_{i=1}^{n} Q_i \delta(x - x_i) \delta(y - y_i) \delta(z - z_i).$$
 (2)

Here,  $Q_i$  represents the emission rate of the *i*th source,  $(x_i, y_i, z_i)$  represent its three-dimensional coordinates, and  $\delta$  is the Dirac delta function. For the rest of this work, x and y will be reserved for describing horizontal coordinates while z will refer to height.

Two important features to note about this equation are its scale invariance and linearity. All the terms on the left-hand side of Equation 1 are linear in C and all of the operators (time derivatives, gradients, dot products, and scalar multiplication) can be distributed across addition. As a result, the solution to the advection-diffusion equation with a set of constant-rate sources can be expressed simply as the sum of the solutions to the partial differential equations associated with each individual source. In other words, the solution to:

$$\frac{\partial C}{\partial t} + \boldsymbol{u}(\boldsymbol{x}, t) \cdot \nabla C - D\nabla^2 C = \sum_{i=1}^n Q_i \delta(x - x_i) \delta(y - y_i) \delta(z - z_i)$$
(3)

can be expressed as  $C = \sum_{i=1}^{n} C_i$  where  $C_i$  is the solution to the advection-diffusion equation applied to the *i*th point source:

$$\frac{\partial C_i}{\partial t} + \boldsymbol{u}(\boldsymbol{x}, t) \cdot \nabla C_i - D\nabla^2 C_i = Q_i \delta(\boldsymbol{x} - \boldsymbol{x}_i) \delta(\boldsymbol{y} - \boldsymbol{y}_i) \delta(\boldsymbol{z} - \boldsymbol{z}_i)$$
(4)

Finally, note that all of the terms on the left-hand side of Equation 4 are linear with respect to C and the operators commute with scalar multiplication: the result of this is that C and  $Q_i$  are directly proportional to one another. Therefore, the solution for an arbitrary emission rate can be obtained by solving this equation once for a unit impulse and then normalizing the concentrations accordingly. This can also be thought of as solving the equation for  $C/Q_i$  and then multiplying it back in the desired rate.

Due to the linear scaling of concentrations with rate and additive nature of discrete point sources, predicted concentrations can

be expressed via a simple linear system that sums up the concentration from every source via b = SQ. Here, b represents a

vector of simulated concentrations, Q is the vector of source rates, and S is the "sensitivity matrix" that describes the transport

of gas from every source to every virtual measurement point. Each row of this matrix corresponds to a simulated methane

measurement at a given time and location under measured meteorological conditions. Each column of the sensitivity matrix

corresponds to a source that is being modeled. The material provided in the Supplemental Information describes how this

sensitivity matrix is calculated for three different dispersion models for later use in the inversion process.

## 3.2 Inversion Frameworks



The primary objective of an inversion framework is to utilize ambient concentration measurements represented as the mass of a pollutant per unit volume of air (e.g., parts-per-million) to estimate the locations and rates of the emission sources as the mass of pollutant per unit time (e.g., kg/hr). The output of dispersion modeling can be expressed as a sensitivity matrix, S, representing the response of every sensor to every potential source. Under the simplifying assumptions of constant emission rates and linear scaling between emission rate and concentration predictions (as implied by Equation 1), inferring the source rates can be expressed as an optimization problem that seeks the vector Q that minimizes an objective function of the residuals:

$$\min_{\mathbf{Q}} f(\mathbf{S}\mathbf{Q} - \mathbf{b}). \tag{5}$$

245 where  $\mathbf{S}Q$  depicts the predicted concentration vector calculated by summing the contribution of all emission sources Q at the measurement locations and times corresponding to the measured concentration vector,  $\mathbf{b}$ .

Selecting an appropriate inversion framework involves balancing computational cost with desired accuracy and control, which all depend on the application's objective. The Supplemental Information presents details on two contrasting options, representing extremes in computational complexity, including a computationally efficient least-squares optimizer along with a more computationally expensive Markov-Chain Monte-Carlo (MCMC) approach. The MCMC inversion method approximates the full posterior distribution function in the high-dimensional parameter space of rate vector Q with more granular control over prior information and the selected objective function.

It should be emphasized that these are only two of many available methods for performing rate inference, which include ge255 netic algorithms, stochastic variational inference, among many others. Rather than implementing an exhaustive list of inversion
solvers, this section aims to apply two example inversion frameworks, spanning a range of complexity, to demonstrate the impact of method selection on the performance of emissions quantification algorithms, as evaluated using several key metrics.

Note that there is no one-size-fits-all "best" framework for this problem. The optimal solution will depend on practical constraints (e.g., computational resources and required latency) and desired outcomes, such as a highly responsive leak detector
that prioritizes detecting emission events of various sizes (even at the cost of false positives), or focusing on accurate estimation of cumulative emissions, even if some smaller leaks are potentially overlooked, or "rolled up" into a smaller number of
larger-rate emission points.

## 3.3 Evaluative Metrics

This study aims to answer the following questions: (i) How well can a CMS, under favorable network configuration conditions (high sensor density) localize emissions to the proper equipment group? (ii) How accurate are the total site-integrated

emissions estimates? (iii) Can an advanced CFD-based forward model, where the wind field is first resolved, outperform the canonical plume and puff models in predicting the concentration field at CMS stations and site-integrated emission rates for these relatively simple controlled release experiments when combined with different inversion frameworks? To this end, the error metrics and evaluation of a given set of rate estimates in comparison to the "ground truth" are tailored to address these specific questions. Although the focus of this study is to investigate the accuracy of different combinations of the forward-inversion frameworks, a more direct comparison of forward models using known emission rates and locations is investigated by computing several statistical error metrics on the predicted and measured concentrations across all forward models in the Supplemental Information.

The result of an individual rate calculation (i.e., a specified dispersion model and inversion framework applied to data from an individual "experiment") will be rate vector Q, where each element of the vector represents the estimated rate associated with equipment group i. For a given experiment, the ground truth rate vector can be equivalently expressed and will be denoted as Q'. For the remainder of this document, any primed values will indicate the actual ground-truth release information, while unprimed values represent the estimated quantities.

## 3.3.1 Localization Metrics






A binary classification scheme is employed to provide a proxy for localization accuracy. In this approach, a given rate vector Q is processed into a binary vector (D) representing the emission status of a given equipment group. If the rate of a given element is 0, then the associated binary element is set to 0 (not emitting), and if the rate is nonzero, the associated binary element is set to 1 (emitting):

$$D_i = \begin{cases} 1 & \text{if } Q_i > 0 \\ 0 & \text{if } Q_i = 0 \end{cases} \tag{6}$$

These binary values are then compared to the ground truth binary values and classified as True Positives (TP), False Positives (FP), False Negatives (FN), and True Negatives (TNs). A TP occurs when both the estimated and actual binary elements are 1 (the equipment group was emitting and properly estimated as emitting). FP indicates that the estimated binary element is 1 but the actual is 0 (an equipment group was estimated to be emitting but was not). FN occurs when the estimated binary is 0 and the actual is 1 (the equipment group was emitting but was not estimated to be emitting), and a TN indicates that both binary elements are 0 (the equipment group was not emitting and was not estimated as emitting). These designations effectively represent the capability of the system to parse out information from the concentration measurements and localize that source to the correct group. For each experiment, the number of correctly identified sources (i.e., the addition of TNs and TPs) is computed to give a localization score (*L*):

$$L = \sum_{j=1}^{M} \begin{cases} 1 & \text{if } D_j = D'_j \\ 0 & \text{if } D_j \neq D'_j. \end{cases}$$
 (7)

A perfect localization score,  $N_{L=5}$ , is achieved when the emission status of each equipment group is correctly identified (as a TN or TP), resulting in a value of L that equals the number of equipment groups in the experiment (the length of Q). In addition, the total number of false positives and false negatives is recorded for each quantification approach (dispersion model plus inversion method) across all experiments. This enables an analysis of each system's tendency to either over-predict or under-predict the number of active sources as a function of dispersion models and inversion frameworks.

## 3.3.2 Quantification Metrics


The metrics in this section are developed to evaluate the accuracy of total site emissions quantification using CMS by comparing estimates to the ground-truth total emissions from the facility. First, total emissions estimates for every single experiment are calculated. The rate vectors for a given experiment are summed before error metrics are computed such that the total emission estimate during an experiment ( $Q_{tot}$ ) is simply:

$$Q_{tot} = \sum_{j} Q_{j}. \tag{8}$$

A set of quantities are then calculated across all experiments. First, the mean error  $(\overline{E})$ , which is a direct measurement of the system's bias (i.e., the mean of the error distribution of facility-level quantified rates) is computed as follows:

$$\overline{E} = \frac{1}{N} \sum_{i=1}^{N} Q_{tot} - Q'_{tot}, \tag{9}$$

where N is the total number of experiments. As a proxy for the uncertainty of the rate estimates, the mean absolute error  $(|\overline{E}|)$  is then calculated. It is a measure of how far off in total the emissions estimates are, on average  $(\pm |\overline{E}| \text{ kg/hr})$ :

$$|\overline{E}| = \frac{1}{N} \sum_{i=1}^{N} |Q_{tot} - Q'_{tot}|$$
 (10)

Analogous quantities (the mean percent error and mean absolute percent error) are computed for the normalized error ( $(Q_{tot} - Q'_{tot})/Q'_{tot}$ ) to better account for low-rate experiments that have less influence on the raw unnormalized error metrics, denoted  $\overline{E}_{rel}$  and  $|\overline{E}|_{rel}$ . Finally, the fraction of rate estimates that are within a factor of two of the actual rate (F2) is computed via:

$$F2 = \frac{1}{N} \sum_{i=1}^{N} \begin{cases} 1 & \text{if } 0.5 

**Figure 3.** Example experiment to illustrate the evaluation of the output of the system with respect to ground truth rates. The image on the left shows each equipment group's estimate classified as either a TP/FP/FN/FP. The upper table summarizes the estimated rates, actual rates, and the detection classification, while the lower table applies the evaluative metrics described above to the data from the upper table.

contribute to the fraction of estimates that were within this factor, when summing over all experiments), and the contribution to the cumulative error from this experiment would simply be  $E\Delta t$ , where  $\Delta t$  is the duration of this experiment. The duration of this particular experiment is 30 minutes, so the contribution to  $\Delta C$  is -0.05 kg.

## 4 Results


Subsection (4.1) details the application of each unique combination of the Puff and Plume dispersion models and inversion framework to the set of 347 experiments. Due to the high computational cost of performing CFD across the entire set of experiments, only a small number of representative cases are computed, the results of which are discussed further in subsection 4.2. The evaluation metrics associated with each combination of Plume/Puff/CFD and LSQ/MCMC are then computed and discussed.

# 4.1 Localization and Quantification Using Gaussian Models

The results obtained by employing each combination of the Gaussian dispersion model and each inversion (LSQ and MCMC) framework across all 347 controlled release experiments are presented in this section. Table 1 details the summary statistics for each combination. In general, the more complex combinations (i.e., puff over plume and MCMC over LSQ) result in better error statistics across the majority of metrics. These improvements are especially prevalent in the localization-related statistics ( $N_{L=5}$  and  $\overline{L}$ ) and the variance of quantification errors (e.g.,  $|\overline{E}|$  and F2). For example, consider the combination of the GPM (Plume) and Least Squares (LSQ) fitting as the simplest combination. In this case, the number of experiments where the emission status of each equipment group are all correctly identified ( $N_{L=5}$ ) is 85 (out of 347). When applying the same LSQ

inversion framework but increasing the complexity of the dispersion model to the Gaussian Puff (Puff), this number goes up to 116. In contrast, holding the dispersion model constant (Plume), but applying the MCMC inversion results in this number increasing to 149. Finally, when using the more sophisticated Puff dispersion model **and** MCMC inversion, the number of cases where all 5 equipment groups' emission status is correctly identified increases to 184.





The improvement in localization statistics when employing the Gaussian Puff instead of the Gaussian plume can be explained by the difference in the fidelity of the temporal modeling of the dispersion. The GPM computes minute-averaged velocity fields, which are assumed to be spatially homogeneous (an assumption that underpins the derivation of the Gaussian Plume) and uses this singular mean value to approximate the dispersion of gas on minute-averaged timescales. In contrast, the Gaussian Puff model directly integrates the spatially and temporally varying wind field on much finer timescales (using 1-Hz wind data), resulting in more accurate dispersion trajectories that take into account the spatial and temporal variation of the wind field.

Improvement in localization statistics when going from the simple LSQ fitting to the MCMC inversion is a direct consequence of the more aggressive sparsity promotion employed in the MCMC algorithm and more sophisticated postprocessing of the iterative 5-dimensional chain of rate vectors and associated probabilities. This results in a significantly smaller number of false positives and also a noticeable decrease in the number of false negatives. This highlights the importance of choosing the appropriate inversion framework for the desired outcome. For instance, if over-localizing (producing nonzero emissions where no emissions were occurring) is not a concern for the application at hand, and only a rough estimate of cumulative emissions is desired, then using the simple LSQ inversion may be appropriate. If, however, the localization output is being used to guide manual detection and remediation efforts (e.g., OGI inspections), then reducing the potential search area via more accurate localization is of critical importance. Therefore, a framework that produces fewer false positives (while maintaining a low false negative count) may be worth the additional computational cost.

Histograms of the number of correctly identified emitters (L) for all 4 combinations of the dispersion model and inversion framework are shown in Figure 4. Note that the number of cases with poor localization results (where only 1 or 2 of the equipment groups' emission status is identified as correct) is significantly lower for the MCMC inversion than the LSQ. Employing a combination of Puff and MCMC results in only 12 (out of 347) experiments with poor localization (localization scores smaller than 3), whereas the combination of Plume and LSQ has 66 cases with poor localization. This highlights the advantage of employing an inversion framework with more aggressive and controllable sparsity promotion. The majority of this improvement is driven by reducing the false positive count, which is achieved by more strictly penalizing nonzero rates.

Figure 5 shows the actual vs estimated facility-integrated rates across all 347 experiments for all 4 combinations of the dispersion model and inversion framework on a logarithmic scale. The black dashed line indicates the parity (y = x) relation.

**Table 1.** Summary statistics of all 4 combinations of dispersion model and inversion calculation.


| Method     | TP  | FP  | FN | TN  | $\overline{E}$ | $ \overline{E} $ | $N_{(L=5)}$ | $\overline{L}$ | F2 | $\overline{E}_{rel}$ | $ \overline{E} _{rel}$ | $\Delta C$ |
|------------|-----|-----|----|-----|----------------|------------------|-------------|----------------|----|----------------------|------------------------|------------|
| Plume LSQ  | 713 | 404 | 62 | 556 | -0.1           | 1.13             | 85          | 3.66           | 73 | 0.13                 | 0.63                   | -135.02    |
| Puff LSQ   | 718 | 319 | 57 | 641 | -0.02          | 0.96             | 116         | 3.92           | 79 | 0.13                 | 0.54                   | -55.89     |
| Plume MCMC | 729 | 242 | 46 | 718 | -0.11          | 0.87             | 149         | 4.17           | 85 | 0.12                 | 0.48                   | -117.44    |
| Puff MCMC  | 743 | 192 | 32 | 768 | 0.02           | 0.8              | 184         | 4.35           | 89 | 0.13                 | 0.42                   | 12.44      |

Figure 4. Histograms of the number of correctly identified emitters across all 4 combinations of dispersion model and inversion calculation.

The bounding dotted black lines show a factor of 2 above and below parity (y = 2x and y = 0.5x) for reference. Note that there is significantly less scatter in the red dots about the dashed black line than there are any other color, indicating that the combination of Puff MCMC yields the tightest distribution of rate estimates about the parity line. This is supported by the F2 statistic from Table 1, which shows that the combination of the Puff MCMC quantification algorithm has the highest percent of estimates within a factor of 2, and the smallest absolute relative error,  $|\overline{E}|_{rel}$ .

Figure 6 shows the actual vs estimated facility-integrated rates across all 347 experiments for various combinations of the dispersion model and inversion framework all together on a linear scale. In this panel, linear fits to the data are shown with the slope and associated  $R^2$  shown in the legend. The linear fits indicate that the quantification estimates that utilized Puff have slopes closer to 1 (0.87 and 0.89 for the Puff LSQ and Puff MCMC, respectively) compared to the quantification estimates that utilized the Plume (0.8 and 0.82 for Plume LSQ and Plume MCMC, respectively). It is worth noting that the slope of this line is not a direct measurement of the bias. This is because these linear fits are generally computed by minimizing the squared error, and as such, a single outlying event with a relatively high error can have an outsized effect on the inferred slope. The slopes

**Figure 5.** Actual vs. estimated facility emissions for 347 experiments on a logarithmic scale. The dashed black line depicts the parity relation (x = y). The dotted lines indicate a factor of 2 lower and higher than the parity relation.

Figure 6. Actual vs. estimated facility emissions for 347 experiments on a linear scale. The dashed black line depicts the parity relation (x = y). The lines correspond to the linear fit to each quantification method's respective actual-vs-estimated pairs. The slopes of these lines and the associated  $R^2$  value are shown in the legend of the right panel.

of these lines are more directly related to the average signed squared error than the bias. This being said, the slopes of these lines are often interpreted as a rough proxy for the bias of a system, and as such, are worth considering with the appropriate context. The trends evident in the linear fits across different quantification estimates are mirrored in the two evaluation metrics that best relate to the bias of the system in Table 1: the average error,  $\overline{E}$ , and the cumulative error,  $\Delta C$ . More specifically, these quantities show the lowest bias (closest to 0 values) for the estimates from the Puff models, which is reflected in Figure 6 (the linear red fit for the quantification estimates using Puff MCMC and orange for the Puff LSQ that have slopes in the parity plots that are the closest to 1).

The coefficient of determination  $(R^2)$  is shown for each linear fit in the right panel of Figure 6. Similar to how the slope, while related, does not directly measure bias,  $R^2$  reflects the variance of the distribution about the linear fit. In other words, it can be used as an indicator for the statistics from Table 1 related to the variance of the error distribution  $(F2, |\overline{E}|, |\overline{E}|_{rel})$ . Similar trends are evident in the  $R^2$  values inferred from the linear fits, with the coefficient of determination of the linear fit getting closer to 1 for increasing complexity in the forward modeling  $(R^2_{Puff} > R^2_{Plume})$ , as well as in the inverse solver  $(R^2_{MCMC} > R^2_{LSQ})$ .


Error histograms for these 4 quantification calculations are shown in Figure 7 to illustrate that the near-zero-error peak is significantly higher for the combination of Puff and MCMC than it is for any of the other quantification methods and drops off more quickly towards higher error. The Plume LSQ results show the most high-error rate estimates, and the Plume MCMC combination generally shows a marginal improvement over the Puff LSQ. The box-and-whisker plots of the relative error distribution associated with each quantification method are presented in the Supplemental Information.

Figure 7. Error histograms for each quantification method across all controlled release experiments.




A pairwise Kolmogorov-Smirnov (KS) test was employed to statistically test the significance of the differences in relative error distributions among the four quantification methods for all 6 combinations of distributions. Table 2 presents the results of the KS statistic and associated p-value for the two error distributions from each combination of quantification methods. The KS statistic represents the maximum difference between cumulative distributions. It can be used as a measure of distribution similarity, with a smaller value indicating greater similarity in distributions. The p-value represents the probability of the two sample distributions being drawn from the same underlying probability distribution. Unsurprisingly, the highest degree of distinction between relative error distributions is observed between Plume LSQ and Puff MCMC methods, the two methods that are respectively identified as the worst and best-performing methods.

The pairwise comparison of Plume LSQ and Puff MCMC error distributions has a KS statistic of 0.17, the highest of any other combination, as well as the lowest p-value of 0.00009. It indicates that the null hypothesis that the two samples could be

- drawn from the same underlying distribution can be rejected to a very high degree of certainty. In contrast, the two LSQ-based inversions have a significantly smaller KS statistic, indicating that the two distributions are more similar than any of the other combinations listed in Table 2. A p-value of 0.208, indicates a substantially higher probability that the two samples could have been drawn from the same underlying distribution.
- It worth noting that 4 out of the 6 combinations of distributions have statistically significant differences (p-values 

**Figure 8.** Scatter plot of Plume LSQ versus Puff MCMC relative errors. Red dots denote experiments where the Puff MCMC outperformed the Plume LSQ calculation, while blue dots denote the opposite.

further setting up the sensitivity matrix, which is the required input for the inversion frameworks. This requires solving five
additional scalar transport equations where for each equation only one of the five emission sources at the ADED facility is
actively emitting at a unit rate. Therefore, quantification and localization estimates using the CFD framework as the forward
model were accomplished for a small subset of experiments. As such, 4 experiments are randomly selected. The CFD framework described in the Supplemental Information is applied to generate sensitivity matrices for each of the 4 experiments. Then,
the inversion process is performed using LSQ and MCMC, and the error metrics described in Section 3.3 are computed. Table
3 highlights the results of the comparison among the three forward models, indicating the performance improvement realized
when employing CFD compared to both Plume and Puff. Only the MCMC results are shown in this table for clarity. Note that
the Plume and Puff results presented in this section are also derived from the same four selected experiments to permit a fair
comparison.

**Table 3.** Table of error metrics across all dispersion models, including CFD, using MCMC inversion applied to the smaller subset of experiments that CFD was performed on.

| Method     | TP | FP | FN | TN | $\overline{E}$ | $ \overline{E} $ | $N_{(L=5)}$ | $\overline{L}$ | F2  | $\overline{E}_{rel}$ | $ \overline{E} _{rel}$ | $\Delta C$ |
|------------|----|----|----|----|----------------|------------------|-------------|----------------|-----|----------------------|------------------------|------------|
| Plume MCMC | 7  | 8  | 1  | 4  | 0.58           | 0.58             | 1           | 2.75           | 1.0 | 0.28                 | 0.28                   | 1.67       |
| Puff MCMC  | 8  | 3  | 0  | 9  | 0.6            | 0.92             | 2           | 4.25           | 1.0 | 0.3                  | 0.45                   | 1.56       |
| CFD MCMC   | 8  | 3  | 0  | 9  | -0.14          | 0.28             | 1           | 4.25           | 1.0 | -0.06                | 0.13                   | -0.61      |

Table 3 indicates comparable localization statistics (TP, FP, FN, TN and  $\bar{L}$ )) for the CFD and Puff models. However, quantifiration statistics for the CFD models show significant improvements over the other dispersion models:  $\bar{E}$  for CFD is -0.14 while the Plume and Puff are 0.58 and 0.6, respectively, a factor of  $\sim 4$  farther from 0. Similarly, the average absolute error,  $|\overline{E}|$  is substantially better (0.28 compared to 0.58 and 0.92 for Plume and Puff, respectively). Relative metrics also show analogous improvements.

Figure 9 shows parity plots for all dispersion models using the MCMC and LSQ inversion frameworks on the left and right, respectively. This figure further emphasizes the performance advantage that CFD-based quantification offers over the Plume and Puff models. While there are some instances where either the Plume or Puff performs better than the CFD, the CFD-based inversion shows an obviously better fit to the parity line than the other methods, on average. As discussed in the Supplemental Information, these improvements result from the CFD approach's ability to reproduce the underlying unsteadiness and the near-surface complex flow effects with greater accuracy and detail.

Figure 9. Parity plots for all three dispersion models using MCMC (left) and LSO (right) for the small sample of CFD-computed experiments

## 485 5 Discussion

The development of multi-source methane emission DLQ algorithms is essential for accurate detection and quantification of oil and gas methane emissions. In these facilities, multiple emissions from different sources, varying in magnitude and location, can occur due to the complex infrastructure and operational processes. The shortcomings of single-source models in disentangling the overlapping plumes from these multiple leaks can lead to significant errors in both the estimated emission rates and the identified leak locations. Multi-source approaches, on the other hand, enable the independent quantification and localization of each individual leak. This capability is crucial for effective facility-level risk assessment and mitigation strategies, as it allows operators to prioritize repairs and address the most significant emission sources. In addition, a comprehensive understanding of the temporal and spatial distribution and magnitude of simultaneous leaks provides a more comprehensive picture of overall site emissions, which is critical for regulatory compliance and accurate emissions inventory development.

490

480

Multi-source methane emission DLQ algorithms require advanced dispersion and inversion methods that account for different aspects of short-range plume transport and inversion. This study represents an initial step toward developing more sophisticated solutions to enable multi-source methane emissions DLQ. However, several simplifying features were implemented in this work, primarily imposed by the data constraints inherent in these specific controlled release studies. Key simplifications include: (i) facility complexity level, (ii) lack of terrain complexity, (iii) lack of complex, time-varying baseline emissions, (iv) constant emission release rates, (v) synchronous emission events during each experiment, and (vi) absence of higher-rate (>10 kg/hr) releases. Furthermore, the focus of this work was on the localization and quantification of constant-rate sources for known emission start and end times: the detection and time-bounding of emission events were not a part of this study.

500

CSU's METEC could be a good representation of relatively simple real-world operational upstream oil and gas facilities. However, other types of facilities, including midstream sites may be more congested, representing an additional complexity level in terms of the number of sources, higher and more fluctuating baseline emissions, emission patterns, and obstructive complexity that may render certain dispersion models inapplicable. Also, the METEC facility is located in an area with fairly simple terrain. However, facilities in other regions with more complex terrain, such as Appalachia, can present other challenges related to natural obstacles. This aspect may require the consideration of alternative dispersion modeling techniques that account for the impact of complex terrain and obstacles, such as the CFD simulations informed by on-site wind measurements presented in this study.

Baseline emissions often depend on many factors, including facility type, site-specific operational activities, facility size, facility age, maintenance practices, and many other considerations. The measurement data collected during the 2024 CSU METEC study did not include any baseline emissions. However, the magnitude of baseline emissions as well as the magnitude of their fluctuations can significantly impact the application of any DLQ solution.

The 2024 CSU METEC study featured only constant emission release rates within each experiment, with simultaneous activation and deactivation of the emitting sources at the experiment's start and end times. Consequently, for each experiment, the
facility alternated between a sterile "off" state and an "on" state with constant emission rates. These simplified and known patterns of emissions constitute "prior" information that algorithms can, in principle, exploit. In addition, the experimental design
required an event-based quantification reporting. This approach may be less practical, and as a result not suited for real-world
applications. In the presence of asynchronously changing time-varying source rates (as expected at operational sites), eventbased quantification will not properly capture the relevant features in emission timeseries. As a result, the 2024 CSU METEC
controlled release testing performance may not fully generalize to the complex emission patterns prevalent in real-world operational settings. It should be noted that to address these concerns, the CSU METEC has developed a more advanced testing
protocol that more accurately replicates the complex emissions found at operational facilities, including simulating operational

background emissions.




The release rates employed for the controlled releases in this study are not sufficiently high for chemical buoyancy to be relevant. However, for large emission events (e.g., super-emitters > 100 kg/hr) neglecting the chemical buoyancy will lead to an overestimation of concentration enhancements for a given source rate, and hence an underestimation of the release rate in the inversion of measurements to rate. Future work with higher rate controlled releases will explore how different approaches to approximating the effects of chemical buoyancy affect the resulting quantification estimates from CMS.

The current study focuses on a small subset of dispersion models and inversion frameworks that are well-established and commonly used for emissions quantification. This deliberate choice was driven by a key objective: to provide a transparent comparison of the performance of methodologies commonly used in atmospheric dispersion modeling and emission quantification. Applying these methods to measurement data with high quality ground-truth releases helps quantify the uncertainty associated with rate estimates. In principle, by applying these same algorithms across different sensor configurations, specific hardware, and sensing modalities (e.g., metal oxide vs. TDLAS), the uncertainties associated with algorithms could be disentangled from the uncertainties inherited from specific deployment strategies and hardware specifications.

In this study, the Gaussian models (plume and puff) were selected for their computational efficiency and widespread application, providing a baseline for comparison. At the opposite end of the complexity spectrum, the CFD modeling was selected for its capability to provide a high-resolution, 3D representation of atmospheric dispersion. This method can capture complex flow patterns that alternative models often overlook. As a result, the CFD modeling allows for detailed simulations of plume behavior, particularly in scenarios involving complex terrain or variable wind fields, where accurate representation of turbulent mixing is crucial. Moreover, CFD models can offer additional benefits to integrate facility-specific data, like site-specific temperature measurements, enabling a more tailored and accurate simulation compared to alternative dispersion models. The MCMC inversion framework was chosen as a more computationally intensive alternative to LSQ for its ability to handle complex, non-linear problems and provide the full high-dimensional posterior probability distribution, which can enable recursive Bayesian estimation for at-scale continuous deployment of these systems (i.e., non-event-based quantification).



Note that the landscape of dispersion modeling and inversion frameworks is far more extensive. The exploration of alternative and often more complex methods, such as Lagrangian stochastic models or more sophisticated computational fluid dynamics (CFD) approaches, could offer valuable insights into the behavior of emissions under complex terrain or highly variable atmospheric conditions. However, for facilities with relatively simple setups and emissions patterns, improvements to the results by employing more sophisticated techniques may be marginal. Therefore, the selection of the most appropriate approach for

quantification depends on the objectives and expected accuracy levels.







While a detailed analysis of how meteorological conditions affect the accuracy of the detection, localization, and quantification algorithms is deferred to a future study, there are several expectations that can be provided based on the underlying physics of gas dispersion and assumptions utilized by Gaussian models that may provide some insight into how they will perform under certain conditions. For instance, the utility of these systems may be significantly decreased during time periods with extremely high wind speeds. This is because the measured concentrations scale inversely with wind speed, so if the wind speed is sufficiently high such that the measured concentration enhancements are within noise of the measurement device, then the system's reliability in terms of converting these concentration enhancements to localized source rates will be significantly inhibited, and in some cases, impossible. The precise wind speed cutoff for this depends on the characteristic source-sensor distances, release rates, atmospheric stability, and the sensitivity of the hardware. In addition to high wind speeds having the potential to negatively impact the performance of CMS-based estimates, extended periods of time with extremely low wind speeds can also pose challenges. In the plume-based implantation presented in the Supplemental Information, measurement data points with corresponding wind speeds of less than 0.5 m/s are excised from the analysis due to the Plume's inapplicability under these circumstances. In practice, this means that if there is a period of time when the wind speed is always less than this threshold, then the plume model, as implemented and presented here, will not be able to quantify emissions from this time period. In contrast, the puff-based model can capture these low wind speed time periods, however the standard dispersion coefficients that are employed may not be as accurate during extremely low-wind speed conditions, when gas pools in place, and as such, the accuracy of puff-based quantification estimates will likely be negatively impacted. Finally, time periods with little variability in wind direction are prone to source confusion (see, e.g., Ball, Eichenlaub, and Lashgari 2025), and as such, the accuracy of the system during these time periods will be negatively impacted. Future work will more quantitatively explore how the output of CMS-based quantification estimates is affected by these various meteorological conditions.

These results represent something of a best-case-scenario in terms of the relative simplicity of the facility as well as the overdense network of sensors that is deployed for this study. In general, the accuracy of the system will likely decrease with lower sensor density. How, exactly, the performance is affected by varying the number of sensors and their configuration will likely depend on the details of the specific facility (number and layout of emission points) as well as the typical variability in the wind direction. In general, we expect the impact of sensor density on DLQ accuracy to be independent of specific model choices (in terms of inverse solvers and forward models). However, future research should explore more quantitatively how the sensor deployment strategy, in terms of both density and configuration, affects the accuracy of various DLQ algorithms from CMS.

Several more in-depth analyses of the performance of these systems and associated and algorithms are possible with this data: generating detection curves as a function of emission rate and inference of 90% detection limits, investigating the per-group detection statistics, rerunning algorithms with different subsets of the underlying sensor data, investigating how well the system

is able to detect small leaks in the presence of larger simultaneous emissions, as well as the impact of emission duration on the DLQ statistics. While the focus of this paper was on some relatively simple evaluative statistics related to the total site-level emissions, and detection and localization accuracy, these more in-depth analyses will be investigated in future work.

It is worth noting that while the puff model, driven by high-frequency spatially-informed wind measurements, outperformed the plume across all metrics, the decision of which model to employ should be informed by the needs of the specific application. For instance: in many cases, high-frequency wind data may not be available, which may render some of the advantages of the puff model moot. Additionally, with the same inversion framework, the plume model's overall quantification estimates were **not** dramatically worse than the puff: for instance, when comparing the Plume+MCMC and Puff+MCMC models, the fraction of estimates within a factor of 2 was only 4% lower, and the mean relative absolute error was only 6% higher when using the plume model. Additionally, the cumulative mass estimate, while showing more negative bias than the puff model, was only off from the true cumulative mass by about 5%. In many cases, such as deployment of these algorithms at scale, especially on facilities without high-frequency wind data (or at extremely simple facilities with no obstructions where the wind field is more homogeneous), the additional computational cost of employing the puff model may not be worth the marginal gains. In cases with more complex wind fields, available high-frequency wind data, and a need for accurate localization, then the puff model should likely be implemented.

Future research should prioritize the evaluation of various quantification methods to refine our understanding and improve the accuracy of emission estimates across diverse operational settings with more complex operational emissions scenarios. This could include more complex facility layouts with a larger number of sources and obstacles, higher baseline emissions with increased fluctuations, more complicated emission patterns (e.g., time-varying emission rates), and case studies located in various regions to account for atmospheric diversity. This highlights the value of conducting more controlled release studies to generate datasets that are representative of various real-world scenarios.

This study underscores one of the primary applications of CMS as long-term integration of emissions for accurate estimates of the total mass emitted. These long-timescale estimates enable the detection of anomalous emission patterns, such as an increased weekly-averaged facility-level emission rate, potentially indicating anomalous events such as a persistent fugitive leak or higher-than-average operational emissions. Future research will examine the impact of sensor density and configuration on quantification accuracy.

## 625 6 Conclusions






This paper presents algorithms for inferring constant-rate emissions from temporally distinct emission events from multiple synchronized emission points. The Gaussian Plume (Plume), Gaussian Puff (Puff), and a CFD-based approach are detailed and coupled with two inversion frameworks, including a simple least-squares estimator with L1 regularization (LSQ) and a Markov Chain Monte Carlo (MCMC) based approach with a spike-and-slab prior. Each combination of the forward model and inversion framework is applied to a set of multi-source constant-rate controlled release experiments. A set of evaluation metrics is presented to investigate the performance of each quantification method for the localization and quantification accuracy. This approach demonstrates the functional dependence of selected key metrics on both forward and inverse modeling techniques.

In general, utilizing Puff, driven by high-frequency 1-Hz wind data and accounting for the spatial inhomogeneity of the wind field results in significantly improved localization statistics when compared to Plume-based estimates. The Puff-based estimates also exhibit a closer-to-zero bias than the Plume-based estimates and reduced variance in the error distribution. Similarly, employing the more sophisticated MCMC-based inversion results in better localization and quantification estimates compared to the simple LSQ fitting. These differences are most stark when comparing the Puff MCMC (most complex) approach to the Plume LSQ (simplest). More specifically, when considering the localization statistics, the number of experiments for which the algorithm correctly identifies the emission status of all of the 5 groups,  $N_{(L=5)}$  in Table 1, increases from 85 to 184 (out of a total of 347 experiments) for the Plume LSQ and Puff MCMC approaches, respectively. Similarly, the mean quantification error improves from -0.1 kg/hr to 0.02 kg/hr, a 5-fold reduction in bias, while the fraction of estimates that were correct to within a factor of 2 (a commonly-used statistic to assess the variance of the error distribution) increases from 73% to 89%.

Under ideal CMS deployment (high sensor density and near-optimal placement), relatively simplified emissions scenarios (constant emission rates, synchronous emissions events during each experiment), and a relatively simple (flat terrain, few obstructions) facility, quantification algorithms applied to data from point sensors can achieve low-bias emissions estimates, leading to accurate long-term estimates of total site emissions. While these systems can achieve near-zero bias, significant uncertainties remain in individual event-based emission rate estimates; the best-performing algorithm studied here (Puff MCMC) still had an average absolute relative error of 42%. Therefore, emissions estimates for any given short timeframe should be interpreted with caution, considering that there is significant uncertainty associated with an individual estimate. However, for cumulative metrics, all models performed reasonably well: as shown in Table 1, the cumulative mass error for the Plume LSQ, Puff LSQ, Plume MCMC, and Puff MCMC were -135, -55, -117, and 12 kilograms, respectively, out of a total of 2,284 kg actually emitted, corresponding to percent errors in cumulative mass estimates of -6%, -2%, -5%, and 0.5%, respectively. This demonstrates that CMS systems, under the conditions present during this testing (sensor deployment and configuration, release rates and patterns, environmental conditions) are capable of highly-accurate cumulative emission estimation, even when using lower-fidelity and simple models such as the Gaussian Plume and simple least-squares based rate inference.

This investigation provides further evidence and confirmation that advanced three-dimensional dispersion modeling approaches, e.g., the large-eddy simulation (LES) type companion CFD numerical experiments carried out in this investigation, can con-

sistently predict more accurate time-varying concentration profiles than plume and puff models across a variety of surface meteorological conditions. This however requires 'nudging' of the momentum and buoyancy transport equations to ensure agreement with the local observations of wind speed and direction. While it may be overly restricting, the spectral profiles presented in the Supplemental Information depict a decent agreement with the spectral content of onsite anemometers. The current results show that while all three models underestimate the concentration field as indicated by FB, NMSE for the CFD model was 25% lower than the puff model. More significantly, the CFD outperformed the puff and plume in 72% and 89% of selected experiments, respectively, in predicting the observed concentration traces. The results obtained via the CFD-based forward model coupled with the two inversion approaches for four of the selected experiments are equally encouraging. As evident in Figure 9, the CFD-based estimates of the inferred emission rates are notably closer to the parity line. Separately, both the absolute and relative error metrics in Table 3 show a marked improvement by the CFD MCMC combination over the other two approaches, with the mean error  $\overline{E}$  of the test-aggregated emission rate showing a near 4-fold improvement. While there is room for improvement as it is a topic of active research and these simulations are certainly more expensive in terms of the computational cost, the CFD results discussed herein offer a proof-of-concept of employing such unsteady tools for use in conjunction with CMS networks on operational and experimental sites. Such advanced tools are expected to find increasing value in setups with numerous obstacles (e.g., power plants and compressor stations), undulating terrain, complex emission profiles, higher release rates, elevated release points and under scenarios where an operational site is only metered by 2-3 continuous monitoring sensors and may not even have an onsite anemometer, thus requiring the use of forecasting tools like the Weather Research and Forecasting (WRF) Model as a surrogate for onsite anemometer(s).




Crucially, the current study demonstrates the significant gains in quantification accuracy achievable with advanced emissions quantification methodologies using fixed-point continuous monitoring systems, particularly for long-timescale cumulative mass estimates, validating their potential for reliable facility-level emissions management.

Data availability. Data for selected experiments is available upon request.

Author contributions. All authors contributed to conceptualizing the project; DRB and UI developed and implemented the models, and analyzed the data; DRB, AL, and UI, wrote the manuscript; all authors revised and edited the manuscript.

Competing interests. The authors declare that they have no conflict of interest.


Acknowledgements. We acknowledge Project Canary for support. We thank the Colorado State University (CSU) Methane Emission Technology Evaluation Center (METEC) for the data collection as part of the Advancing Development of Emissions Detection (ADED) project, funded by the Department of Energy's National Energy Technology Laboratory (NETL).

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
