# Peer review of "Performance Evaluation of Multi-Source Methane Emission Quantification Models Using Fixed-Point Continuous Monitoring Systems"

_EGUsphere, 2025_

## Author Comment (AC1)

**Response to the Referees' Comments**

**REFEREE #1**

**Greenhouse gas inventories and facility operations – Could the authors elaborate on how this technique will aid facilities in improving their greenhouse gas inventories? It is mentioned in the last line of the abstract, but I'd like to see a paragraph on the application of these methods to oil and gas facility operators, how it could be implemented, and how easy would the models be for operators to use?**

**Authors:**

Thanks for your comment. We added additional wording per Referee's comment.

**Changes:**

Accurate emissions quantification using CMS can enhance the reliability and robustness of GHG emissions inventory development. Traditional inventory methods often rely on activity data and generic emission factors, which fail to capture the dynamic nature of emissions from individual sources or facilities. By providing continuous, real-time measurements source-specific emissions, CMS offers a direct and empirically driven approach to quantify actual emissions. High temporal resolution of the CMS measurement allows for the identification and characterization of gas releases, including the duration and frequency of emission events. In addition, accurate quantification offers a more in-depth understanding of the magnitude of fugitive emissions, intermittent events, and variations in operational performance that are often missed by periodic or estimation-based methods. Integrating CMS data into GHG inventories leads to a more comprehensive understanding of emission sources, enables the tracking of emission reduction efforts with greater confidence, and supports the development of more granular and verifiable inventories, informing climate policies, and tracking progress towards decarbonization goals.

**Length/brevity – The authors clearly know these models well and have robust statistical evidence for the accuracy of the models, but this paper is around 50 pages long. I would like to see the authors work on making the paper more concise. Perhaps some of the introduction can be cut or combined to make the thesis of the paper clearer. Also, maybe some parts of Section 3 Methodology can be moved to a supplement, and the detailed explanation of the summary statistics to the supplement (keeping some of the explanation in the main body obviously). Right now, the introduction and methodology read like a literature review, so I'd like to see it condensed to focus on the main point of the paper that the Puff/MCMC model works best for the CM system and the data explaining why. Another suggestion could be to**

**add the description of the figures to the figure captions themselves instead of in the body of the paper.**

**Authors:**

We moved parts of the Introduction and Methodology sections to the Appendix to make the paper more concise.

**Changes:**

Please see Appendices A and B.
* * *
**Meteorological Conditions – I'd like to see in the discussion how the meteorological conditions could affect the outcome of the various models, for example if it's very windy would more leaks be missed? How does meteorology affect the performance of the CMS emissions models?**

**Authors:**

Thanks for your comment. We have added a paragraph in the discussion about how certain meteorological conditions may affect the performance of the system, but defer a more in-depth analysis of accuracy as a function of various conditions to future work, as the paper is already long to the point of being somewhat cumbersome, as pointed out by both reviewers.

**Changes:**

While a detailed analysis of how meteorological conditions affect the accuracy of the detection, localization, and quantification algorithms is deferred to a future study, there are several expectations that can be provided based on the underlying physics of gas dispersion and assumptions utilized by Gaussian models that may provide some insight into how they will perform under certain conditions. For instance, the utility of these systems may be significantly decreased during time periods with extremely high wind speeds. This is because the measured concentrations scale inversely with wind speed, so if the wind speed is sufficiently high such that the measured concentration enhancements are within noise of the measurement device, then the system's reliability in terms of converting these concentration enhancements to localized source rates will be significantly inhibited, and in some cases, impossible. The precise wind speed cutoff for this depends on the characteristic source-sensor distances, release rates, atmospheric stability, and the sensitivity of the hardware. In addition to high wind speeds having the potential to negatively impact the performance of CMS-based estimates, extended periods of time with extremely low wind speeds can also pose challenges. In the plume-based implantation presented in Section 3.1.1, measurement data points with corresponding wind speeds of less than 0.5 m/s are excised from the analysis due to the Plume's inapplicability under these circumstances. In practice, this means that if there is a period

of time when the wind speed is always less than this threshold, then the plume model, as implemented and presented here, will not be able to quantify emissions from this time period. In contrast, the puff-based model can capture these low wind speed time periods, however the standard dispersion coefficients that are employed may not be as accurate during extremely low-wind speed conditions, when gas pools in place, and as such, the accuracy of puff-based quantification estimates will likely be negatively impacted. Finally, time periods with little variability in wind direction are prone to source confusion (see, e.g., Ball, Eichenlaub, and Lashgari 2025), and as such, the accuracy of the system during these time periods will be negatively impacted. Future work will more quantitatively explore how the output of CMS-based quantification estimates is affected by these various meteorological conditions.

**Paper thesis - I'd like to see more discussion on why the Puff model is the most reasonable approach operationally. Is the point of the paper to compare different measurement techniques or to offer a solution for facilities to implement?**

**Authors:**

Thanks for your comment. We have added a paragraph in the discussion that goes into a bit more detail on the choice of models and the factors that may be in play (computational cost, complex wind fields, availability of high-frequency wind data, at-scale deployment) when deciding on which model to deploy.

**Changes:**

It is worth noting that while the puff model, driven by high-frequency spatially-informed wind measurements, outperformed the plume across all metrics, the decision of which model to employ should be informed by the needs of the specific application. For instance: in many cases, high-frequency wind data may not be available, which may render some of the advantages of the puff model moot. Additionally, with the same inversion framework, the plume model's overall quantification estimates were not dramatically worse than the puff: for instance, when comparing the Plume+MCMC and Puff+MCMC models, the fraction of estimates within a factor of 2 was only 4% lower, and the mean relative absolute error was only 6% higher when using the plume model. Additionally, the cumulative mass estimate, while showing more negative bias than the puff model, was only off from the true cumulative mass by about 5%. In many cases, such as deployment of these algorithms at scale, especially on facilities without high-frequency wind data (or at extremely simple facilities with no obstructions where the wind field is more homogeneous), the additional computational cost of employing the puff model may not be worth the marginal gains. In cases with more complex wind fields, available high-frequency wind data, and a need for accurate localization, then the puff model should likely be implemented.

**Cumulative facility emissions rate – Could you elaborate on the sentence in the conclusion on the cumulative mass emissions estimate (L947)? I think this is critical for understanding total flux emissions from a facility and deserves more attention. Maybe a table comparing all the methods together and how they compare them to what METEC reported.**

**Authors:**

Thanks for your comment. We have added several sentences clarifying the interpretation of the cumulative mass estimate error shown in Table 1 across the different models.

**Changes:**

However, for cumulative metrics, all models performed reasonably well: as shown in Table 1, the cumulative mass error for the Plume LSQ, Puff LSQ, Plume MCMC, and Puff MCMC were -135, -55, -117, and 12 kilograms, respectively, out of a total of 2,284 kg actually emitted, corresponding to percent errors in cumulative mass estimates of

-6%, -2%, -5%, and 0.5%, respectively. This demonstrates that CMS systems, under the conditions present during this testing (sensor deployment and configuration, release rates and patterns, environmental conditions) are capable of highly-accurate cumulative emission estimation, even when using lower-fidelity and simple models such as the Gaussian Plume and simple least-squares based rate inference.

**Validation – could you elaborate more on validating the models against the reported METEC emissions and locations? (goes with the previous comment)**

**Authors:**

Thanks for your comment. We have added a case study experiment to illustrate an example experiment, associated release rates, estimates, and evaluation to help clarify the meaning of some of these metrics

**Changes:**

To aid in the understanding of these metrics, an illustrative example of these evaluative metrics applied to a single experiment from the study is shown in Figure 3. This figure shows an image and two tables that summarize the output of the system (rate estimates for each equipment group) alongside the ground-truth release rates (top table) and computes the relevant evaluative metrics for this single experiment (bottom table). During this experiment, there were three active release sources: the tanks (group 4T), the western separators (group 4S), and the eastern separators (group 5S). The western and eastern wellheads (groups 4W and 5W, respectively) were not emitting. The quantification estimates are shown in the 2nd column of the top table, while the ground-truth release rates are shown in the 3rd column of the top table. The final column shows the classification of the estimate as either a TP/FP/FN/FP as previously described. The table

on the bottom shows the relevant evaluative metrics applied to the estimated and ground-truth rates in the top table. We see that, for this example experiment, there were 2 true positives (the system accurately identified that both the 4S and the 5S groups were emitting), one false negative (the system missed that the tanks were emitting), one false positive (the system assigned a small but nonzero rate to the 4W group, which was not emitting), and one true negative (the system accurately identified that the 5W group was not emitting). These statistics are summarized in the bottom table, along with the overall "localization score", which in the case, was 3 (i.e., the emission status of 3 out of the 5 equipment groups were correctly identified). The total estimated and actual facility-level emission rate is shown in the bottom table as $Q$ and $Q'$ (these are computed as the sum down the "Estimated Rate" and "Actual Rate" columns, respectively). In this example, the estimated facility rate is 1.73 kg/hr while the actual emission rate is 1.83, representing an error of -0.1 kg/hr ($E$) and a relative error of -0.055 (i.e., -5.5% error, $E_{rel}$). In terms of the other quantification-related metrics ($F2$ and $\Delta C$), this experiment's estimated facility-level rate is within a factor of 2 of the actual rate (so it would positively contribute to the fraction of estimates that were within this factor, when summing over all experiments), and the contribution to the cumulative error from this experiment would simply be $E.\Delta t$, where $\Delta t$ is the duration of this experiment. The duration of this particular experiment is 30 minutes, so the contribution to $\Delta C$ is -0.05 kg.

[Figure]

| Group | Estimated Rate | Actual Rate | Classification |
|---|---|---|---|
| 4S | 0.26 | 0.52 | TP |
| 4T | 0.0 | 0.81 | FN |
| 4W | 0.05 | 0.0 | FP |
| 5S | 1.42 | 0.5 | TP |
| 5W | 0.0 | 0.0 | TN |

| TP | FP | TN | FN | L | Q | Q' | E | $E_{rel}$ |
|---|---|---|---|---|---|---|---|---|
| 2.0 | 1.0 | 1.0 | 1.0 | 3.0 | 1.73 | 1.83 | -0.1 | -0.055 |

Figure 3. Example experiment to illustrate the evaluation of the output of the system with respect to ground truth rates. The image on the left shows each equipment group's estimate classified as either a TP/FP/FN/FP. The upper table summarizes the estimated rates, actual rates, and the detection classification, while the lower table applies the evaluative metrics described above to the data from the upper table.

**Number of CMS instruments – How many CMS instruments would be needed to perform an accurate DLQ using the Puff/MCMC method (and other methods as well)?**

**Authors:**

Thanks for your comment. We have added a paragraph in the discussion on the expected impact of sensor density on DLQ accuracy.

**Changes:**

These results represent something of a best-case-scenario in terms of the relative simplicity of the facility as well as the overdense network of sensors that is deployed for this study. In general, the accuracy of the system will likely decrease with lower sensor density. How, exactly, the performance is affected by varying the number of sensors and their configuration will likely depend on the details of the specific facility (number and layout of emission points) as well as the typical variability in the wind direction. In general, we expect the impact of sensor density on DLQ accuracy to be independent of specific model choices (in terms of inverse solvers and forward models). However, future research should explore more quantitatively how the sensor deployment strategy, in terms of both density and configuration, affects the accuracy of various DLQ algorithms from CMS.

**Introduction – Add references for factual statements L16, L19, L26, L43 (AVO), L45 (OGI), L73 ('smoke alarms'), L75 (CMS development), L859 (why Appalachia?)**

**Authors:**

Thanks for your comment. We added references to the referred parts of the Introduction.

**Changes:**

Please see the updated Introduction section.

**L12 – "anomalous (emissions) patterns"?**

**Authors:**

We added "emission" to the abstract

**Changes:**

The study highlights the importance of long-term integration for accurate total mass emission estimates and the detection of anomalous emission patterns.

**L55 – Which one, 1-2 kg/hr or 200 kg/hr?**

**Authors:**

We edited the text to clarify this point.

**Changes:**

Satellite and aerial remote sensing techniques can detect emissions from specific sources, with aerial methods capable of detecting emissions as low as 1-3 kg/hr, while satellites have minimum detection limits of to approximately 200 kg/hr or higher (Sherwin, Rutherford, Chen, et al. 2023).

**The last two paragraphs of the intro are good. More of this.**

**Authors:**

We added three paragraphs to the intro, covering key research questions, the role of accurate quantification in developing emission inventories, and the use of controlled release data in this study.

**Changes:**

Three key questions will be addressed in this study: (i) under an optimum sensor density and placement, how effectively can a CMS pinpoint emissions to the correct equipment group? (ii) what is the accuracy of the total site-integrated emissions estimates for such CMS network? And, (iii) How well can an advanced CFD-based forward model, coupled with various inversion frameworks perform in predicting emission rates compared to traditional plume and puff models?

Accurate emissions quantification using CMS can enhance the reliability and robustness of GHG emissions inventory development. Traditional inventory methods often rely on activity data and generic emission factors, which fail to capture the dynamic nature of emissions from individual sources or facilities. By providing continuous, real-time measurements source-specific emissions, CMS offers a direct and empirically driven approach to quantify actual emissions. High temporal resolution of the CMS measurement allows for the identification and characterization of gas releases, including the duration and frequency of emission events. In addition, accurate quantification offers a more in-depth understanding of the magnitude of fugitive emissions, intermittent events, and variations in operational performance that are often missed by periodic or estimation-based methods. Integrating CMS data into GHG inventories leads to a more comprehensive understanding of emission sources, enables the tracking of emission reduction efforts with greater confidence, and supports the development of more granular and verifiable inventories, informing climate policies, and tracking progress towards decarbonization goals.

In this study selected quantification algorithms are evaluated using the data from controlled release experiments featuring constant-rate emission events with known start and end times. However, it's crucial to recognize that these controlled release scenarios are highly idealized, as they involve constant release rates and simultaneous emissions from all active sources. This idealization may impact the practical applicability of these algorithms in more complex, real-world conditions. A more in-depth evaluation of the performance of fixed-point CMS in complex emission environments is provided in a separate study (Ball, Eichenlaub, and Lashgari 2025).

**Data – Conceptually, I'm having a hard time understanding how a controlled release works, are there only 5 locations where a leak could be? Does each leak have a different release rate? Could my confusion be cleared up by a more detailed figure?**

**Authors:**

Thanks for your comment. We added more wording to explain it.

**Changes:**

This paper aims to address the critical need for developing a more comprehensive understanding of the performance and robustness of various multi-source methane quantification methods by evaluating the performance of several established atmospheric dispersion modeling and inversion frameworks within a controlled, multi-leak experimental setting with synchronous emission sources and constant rates.

**L205 – This is good and targets the scope of the paper, move this to the intro?**

**Authors:**

Thanks for your comment. We added a paragraph with similar content to the Intro.

**Changes:**

In this study selected quantification algorithms are evaluated using the data from controlled release experiments featuring constant-rate emission events with known start and end times. However, it's crucial to recognize that these controlled release scenarios are highly idealized, as they involve constant release rates and simultaneous emissions from all active sources. This idealization may impact the practical applicability of these algorithms in more complex, real-world conditions. A more in-depth evaluation of the performance of fixed-point CMS in complex emission environments is provided in a separate study (Ball, Eichenlaub, and Lashgari 2025).

**Figure 2 – Make the wind data into a wind rose? Use lat/lon for the location figure instead of x and y? Are the colors in the concentration figure for each instrument?**

**Authors:**

Thanks for your comment. We added individual anemometer measurements to bottom right panel as opposed to mean values and added wording to clarify the meaning of colors and coordinate system that facility layout is shown in.

**Changes:**

Figure 2 offers a visual illustration of the layout of the controlled release facility (left), including bounding boxes around each of the 5 equipment groups (left) and sensor locations (x's). It also shows measurement data from a randomly selected controlled release experiment, including concentration measurements from individual sensors (top right) and the QU and V components of the anemometer measurements (with solid and dotted lines, respectively, bottom right). The colors of the curves in the right panels here correspond to the colored x's in the left panels. This figure encapsulates all of the data necessary to run quantification algorithms (sensor locations, source locations, concentration timeseries data, and wind timeseries data).

[Figure]

**L290 – How do you determine k? Is it a measurement?**

**Authors:**

Thanks for your comment. We added a sentence clarifying the significance of the wavenumber, k.

**Changes:**

It is important to note how different turbulent wavenumbers (k) affect a plume at different characteristic scales (L). For reference, k can be thought of simply as the inverse spatial scale of a turbulent eddy, k = 2π/$L_{eddy}$ , where Leddy represents the characteristic length scale associated with a particular turbulent eddy.

**L317 – remove quotes from Pasquill, it is a noun (person).**

**Authors:**

We removed the quotes.

**Changes:**

Many of the standard methods for computing the dispersion coefficients $\sigma_y$ and $\sigma_z$ rely first on an approximation of the Pasquill atmospheric stability class (ASC).

**L345 – Could you elaborate on which 'fewer assumptions' are made in Gaussian Puff?**

**Authors:**

We have reworded the introductory paragraph of the gaussian puff to more clearly link its capabilities to the underlying assumptions (or lack of assumptions).

**Changes:**

The Gaussian Puff model is a Lagrangian approach to approximating the solution to the advection-diffusion equation that makes fewer assumptions than the Gaussian Plume. More specifically, this method can capture the relevant physical effects embedded in spatially varying wind fields (i.e., it does not assume homogeneous wind fields), can handle time-varying emission rates (does not assume steady-state emission rate), and also more properly account for low wind speeds and unstable conditions when the wind vector rapidly changes (does not assume steady-state wind fields).

**Section 3.3 – I like the beginning of the first paragraph that explains the questions this study is attempting to answer – could you make this clearer in the introduction? Maybe move this to the intro?**

**Authors:**

Thanks for your comment. We added a paragraph with similar content to the Intro.

**Changes:**

Three key questions will be addressed in this study: (i) under an optimum sensor density and placement, how effectively can a CMS pinpoint emissions to the correct equipment group? (ii) what is the accuracy of the total site-integrated emissions estimates for such CMS network? And, (iii) How well can an advanced CFD-based forward model, coupled with various inversion frameworks perform in predicting emission rates compared to traditional plume and puff models?

**Figure 8,9 – Move to supplement?**

**Authors:**

Thanks for your comment. We moved Figure 9 as well as the text associated with it to the Appendix. However, we believe that Figure 8 is a key part of the results, helping readers better understand the differences between various methods. As a result, we would like to keep it in the main body.

**Changes:**

Please see Appendix D.

---

## Author Comment (AC2)

**Response to the Referees' Comments**

**Half the metrics are directly related to detection (TP, FP, FN, TN, N(L=5), Lbar), however how is detection actually done in the different methods?**

**Authors:**

Thanks for your comment. We have added a case study experiment to illustrate an example experiment, associated release rates, estimates, and evaluation to help clarify the meaning of some of these metrics.

**Changes:**

To aid in the understanding of these metrics, an illustrative example of these evaluative metrics applied to a single experiment from the study is shown in Figure 3. This figure shows an image and two tables that summarize the output of the system (rate estimates for each equipment group) alongside the ground-truth release rates (top table) and computes the relevant evaluative metrics for this single experiment (bottom table). During this experiment, there were three active release sources: the tanks (group 4T), the western separators (group 4S), and the eastern separators (group 5S). The western and eastern wellheads (groups 4W and 5W, respectively) were not emitting. The quantification estimates are shown in the 2nd column of the top table, while the ground-truth release rates are shown in the 3rd column of the top table. The final column shows the classification of the estimate as either a TP/FP/FN/FP as previously described. The table on the bottom shows the relevant evaluative metrics applied to the estimated and ground-truth rates in the top table. We see that, for this example experiment, there were 2 true positives (the system accurately identified that both the 4S and the 5S groups were emitting), one false negative (the system missed that the tanks were emitting), one false positive (the system assigned a small but nonzero rate to the 4W group, which was not emitting), and one true negative (the system accurately identified that the 5W group was not emitting). These statistics are summarized in the bottom table, along with the overall "localization score", which in the case, was 3 (i.e., the emission status of 3 out of the 5 equipment groups were correctly identified). The total estimated and actual facility-level emission rate is shown in the bottom table as $Q$ and $Q'$ (these are computed as the sum down the "Estimated Rate" and "Actual Rate" columns, respectively). In this example, the estimated facility rate is 1.73 kg/hr while the actual emission rate is 1.83, representing an error of -0.1 kg/hr ($E$) and a relative error of -0.055 (i.e., -5.5% error, $E_{rel}$). In terms of the other quantification-related metrics ($F2$ and $\Delta C$), this experiment's estimated facility-level rate is within a factor of 2 of the actual rate (so it would positively contribute to the fraction of estimates that were within this factor, when summing over all experiments), and the contribution to the cumulative error from this experiment would simply be $E.\Delta t$, where $\Delta t$ is

the duration of this experiment. The duration of this particular experiment is 30 minutes, so the contribution to ΔC is -0.05 kg.

[Figure]

| Group | Estimated Rate | Actual Rate | Classification |
|---|---|---|---|
| 4S | 0.26 | 0.52 | TP |
| 4T | 0.0 | 0.81 | FN |
| 4W | 0.05 | 0.0 | FP |
| 5S | 1.42 | 0.5 | TP |
| 5W | 0.0 | 0.0 | TN |

| TP | FP | TN | FN | L | Q | Q' | E | $E_{rel}$ |
|---|---|---|---|---|---|---|---|---|
| 2.0 | 1.0 | 1.0 | 1.0 | 3.0 | 1.73 | 1.83 | -0.1 | -0.055 |

Figure 3. Example experiment to illustrate the evaluation of the output of the system with respect to ground truth rates.  The image on the left shows each equipment group's estimate classified as either a TP/FP/FN/FP.  The upper table summarizes the estimated rates, actual rates, and the detection classification, while the lower table applies the evaluative metrics described above to the data from the upper table.

A lot of space spent on the models and metrics, comprising Section 3, a lot of which is already described in the literature. I think it could be shortened, if possible, to better highlight the results Section 4. A key novelty of the manuscript is the multi-source estimation estimate, along with large number of experiments with continuous monitors. Many analyses could be envisioned, in particular detection curve vs emission rate, whether any equipment groups perform better (perhaps due to prevailing wind patterns or other factors), simulating if there were fewer sensors (as mentioned might be realistic), interference (if small leaks sometimes are hidden by larger ones), effect of experiment time vs DLQ accuracy (30 minutes vs 8 hours), etc.

**Authors:**

We moved parts of the Introduction and Methodology sections to the Appendix to make the paper more concise.

**Changes:**

Please see Appendices A and B.

**L43 add AVO in parentheses**

**Authors:**

We added the abbreviation in parentheses.

**Changes:**

Traditional approaches for detecting methane emissions often rely on human senses (auditory, visual, and olfactory (AVO) inspections) or portable sensors used in close proximity to potential sources.

**Cheptonui 2024 a/b are same paper**

**Authors:**

We corrected the reference.

**Changes:**

Several studies have independently evaluated the efficacy of CMS in quantifying emissions, suggesting promising advancements in recent years (Bell et al. 2023; Ilonze et al. 2024; Cheptonui et al. 2025).

More details on the data collected as part of the 2024 CSU METEC controlled release study can be found elsewhere (Cheptonui et al. 2025).

**Sec 2 – for releases at multiple equipment groups, does each release rate simply randomly belong to the overall distribution in Fig 1b?**

**Authors:**

That is correct. We have added a figure and some explanation that more concretely shows the releases from a single experiment, the output of the system, and how this particular case is "scored".

**Changes:**

The experiments are designed such that only one release point is active per equipment group at the METEC facility. Each equipment group is composed of numerous "equipment units" (i.e., individual tanks, wellheads, or separators) and each equipment unit may have multiple potential release points on it. In other words, each equipment group has numerous \textit{potential} release points, but only one is ever active at a time for a given experiment. In this study, we focus on the ability of the system to correctly detect, localize, and quantify to the equipment group level. As such, the centroid of each equipment group is computed and these 5 coordinate pairs (corresponding to the 5 equipment groups at the facility) are used as the potential source locations as an input to the localization and quantification (LQ) algorithms.

**L270 I am surprised to characterize the importance of the stability class and dispersion coefficient parameterization as minimal. Sure they may be representing the same behavior, but don't they empirically have a significant effect?**

**Authors:**

To clarify, the stability class and dispersion coefficients indeed play a large role in predicting concentrations (and hence, quantified flux rates). However, many of the parametrizations of them are simply different empirical formulae, derived from the same underlying publicly available data, and give very similar approximations of the stability class and dispersion as a function of distance. If a poor parametrization is used, then the performance of the quantification will be poor. However, most of the commonly accepted and standard methods are very similar to one another and are just different functional forms and associated coefficients. We have added a sentence clarifying this statement

**Changes:**

The following subsections provide brief overviews of the theory underpinning the dispersion models, followed by more specific implementation details. Note that there are myriad small choices (e.g., stability class calculations, dispersion parametrization) that must be made in the data processing and algorithmic workflow when it comes to running these dispersion models. It is outside the scope of this study to enumerate and present results from every combination of valid choices. Instead, we will provide clear justifications for the specific choices made in this study and demonstrate the efficacy of the models under these specific implementations. It should be noted that the impact of most of these higher-order decisions on the results is minimal, as they are often different approaches of approximating the same underlying phenomena. For example, there are several commonly-used functional forms and associated coefficients to describe how the dispersion of a gas plume scales with distance. While these empirical formulae may look very different (e.g., some utilize power laws while others employ logarithms), they are generally inferred by fitting these functional forms to the same underlying data, and result in similar general characteristics despite the sometimes dramatically different functional forms.
* * *
**L275 GMP -> GPM**

**Authors:**

Thanks for your note. We corrected the error.

**Changes:**

The commonly used Gaussian Plume model (GPM) provides a closed-form solution to the steady-state advection-diffusion equation for a single point source emitting at rate Q from height H.
* * *
**Eq 8 (for the Gaussian puff) seems unusual – please check. Normally there is a 2\*pi^(3/2) factor, and importantly, time dependence**

**Authors:**

Thanks for your comment. The correct exponent of 3/2 on the 2*pi has been added to the equation, and the explicit time dependence is mentioned.

**Changes:**

$$C(x,y,z,t) = \frac{Q\Delta t}{(2\pi)^{3/2}\sigma_x\sigma_y\sigma_z}\exp\left[-\frac{y^2}{2\sigma_y^2} - \frac{(z-H)^2}{2\sigma_z^2} - \frac{(z+H)^2}{2\sigma_z^2} - \frac{x^2}{2\sigma_x^2}\right]. \tag{8}$$

**L439 Is a no flux condition typical in these simulations? A 200 m boundary seems like it would significantly affect the dispersion behavior**

**Authors:**

Thanks for your note. If we were considering much larger length scales, then the 200m zero-flux upper boundary may artificially impact on the results, however here, the average source-sensor distance is around 50 meters, and the releases are effectively at ground level. As such, virtually no simulated gas reaches the "ceiling" of the simulation within the relevant length scales of source->sensor.

**Changes:**

No changes were made.

**Fig 5 some information is not visible on this plot (puff LSQ on L=4 and plume MCMC on L = 1)**

**Authors:**

Thanks for your note. The plot has been modified to include variable line widths and marker styles so that overlapping lines are more distinguishable.

**Changes:**

[Figure]

**Figure 11 suggest using different marker types, as the colors by themselves can be difficult to distinguish**

**Authors:**

Thanks for your note. Each model now has a unique marker in the plot and the marker size has been increased to help with visibility.

**Changes:**

[Figure]

**L173 Inverse distance weighting is mentioned, and "sonic anemometers" (plural) here, but is unclear to me where/how many where used. Could this be added to Fig 2 or otherwise?**

**Authors:**

Thanks for your note. We have added a sentence clarifying that each of the sensors in this study was equipped with an anemometer in Section 2 and also added some wording around this in discussing the inverse distance weighting.

**Changes:**

Ten Canary X integrated devices were installed within the METEC site perimeter to measure ambient methane concentrations. All of the Canary X devices used for this study were equipped with an anemometer.

---

## Referee Report (RR1)

**Round 2, Reviewer 1 Comments:**

"Performance Evaluation of Multi-Source Methane Emissions Quantification Models Using Fixed-Point Continuous Monitoring Systems"

July 3rd, 2025

**Dear Authors,**

Thank you for taking the time to incorporate my comments into this draft of your paper. The science questions I had earlier have all been addressed well. I still have a few general comments on the writing and editing of the paper that I would like to see improved by the next draft round:

- The introduction is much better and more concise now, good job! I don't think you need Appendix A at all.
- Recommend making the following sections in "Data" for the paragraphs that already exist: "Measurements" for L130-144, "Field Site" for L146-153, and "Controlled Releases" for L155-183.
- I'm confused by the appendices. I think they should just be moved to a supplement.
- I still feel that the methods section is too long and over-explanatory. I echo reviewer 2's comment that many of the explanation in the methodology is already described in the literature. Keep moving more to the supplement and remain focused on the key point of the paper, i.e. the results and discussion sections. You can have 1-2 paragraphs explaining the model, then refer the reader to the supplement to learn more specific things about the model.
- The results and discussion are also very long and could be edited down to fewer and shorter paragraphs.
- Style/Structure: I'd really like to see the writing itself edited down to reduce the wordy-ness. You could cut at least 20% of the text and still communicate your ideas well. Keep sentences shorter and concise. Pare down the paragraphs to focus on a main point. An editor would be useful.

I look forward to reading the next draft of your paper.

---

## Author Response (AR2)

**Dear Dr. Presto,**

Thank you for the decision to "Publish subject to minor revisions" for our manuscript. We have carefully reviewed and implemented the specific minor revisions requested. We confirm that both issues have been fully addressed:

- 1. We moved appendices to a separate Supplemental Information
- 2. We have significantly shortened the version for final publication by moving the content related to dispersion models and inverse frameworks to the SI.

We believe that these revisions, alongside the previous reviewer comments we addressed, have improved the manuscript. We look forward to your final decision.

**Sincerely,**

Ali Lashgari, PhD Manager of Scientific & Academic Partnerships Project Canary, PBC

(304) 777-9760 ali.lashgari@projectcanary.com 1200 17th St., 23rd floor, Denver, CO, 80202 LinkedIn